



# Evaluating and improving the treatment of gases in radiation schemes: the Correlated K-Distribution Model Intercomparison Project (CKDMIP)

Robin J. Hogan[1] and Marco Matricardi[1]

[1]European Centre for Medium-range Weather Forecasts, Reading, UK.

**Correspondence:** Robin J. Hogan (r.j.hogan@ecmwf.int)

**Abstract.** Most radiation schemes in weather and climate models use the 'correlated $k$-distribution' (CKD) method to treat gas absorption, which approximates a broadband spectral integration by $N$ pseudo-monochromatic calculations. Larger $N$ means more accuracy and a wider range of gas concentrations can be simulated, but at greater computational cost. Unfortunately, the tools to perform this efficiency–accuracy trade-off (e.g., to generate separate CKD models for applications such as short-range

weather forecasting to climate modelling) are unavailable to the vast majority of users of radiation schemes. This paper describes the experimental protocol for the Correlated K-Distribution Model Intercomparison Project (CKDMIP), whose purpose is to use benchmark line-by-line calculations: (1) to evaluate the accuracy of existing CKD models, (2) to explore how accuracy varies with $N$ for CKD models submitted by CKDMIP participants, (3) to understand how different choices in way that CKD models are generated affects their accuracy for the same $N$, and (4) to generate freely available datasets and software facilitat-

ing the development of new gas-optics tools. The datasets consist of the high-resolution longwave and shortwave absorption spectra of nine gases for a range of atmospheric conditions, realistic and idealized. Thirty-four concentration scenarios for the well-mixed greenhouse gases are proposed to test CKD models from palaeo- to future-climate conditions. We demonstrate the strengths of the protocol in this paper by using it to evaluate the widely-used Rapid Radiative Transfer Model for General Circulation Models (RRTMG).

**1  Introduction**

The gas absorption spectra of planetary atmospheres typically contain hundreds of thousands of spectral lines, so line-by-line radiative transfer calculations require O($10^6 - 10^7$) monochromatic calculations to cover the full shortwave and longwave spectrum, which is far too costly for most applications. The correlated $k$-distribution (CKD) method (e.g., Goody et al., 1989; Lacis and Oinas, 1991; Petty, 2006) avoids the need to resolve spectral lines by reordering the mass absorption coefficient,

$k(\nu)$, over a particular range of wavenumbers, $\nu$, such that the resulting function $k(g)$ increases smoothly and monotonically from the least absorbing ($g = 0$) to the most absorbing ($g = 1$). The smooth function $k(g)$ may be discretized using far fewer quadrature points than $k(\nu)$, with the result that the entire shortwave and longwave spectrum can be represented by O($10^2$) independent pseudo-monochromatic calculations, usually referred to as $k$ *terms* or $g$ *points*. In order to perform radiation





calculations over the full atmospheric column, we typically need to assume perfect rank correlation between the $k$ spectra at
each height. The CKD method has the advantage over random-band models that it is easy to incorporate scattering.

The more $k$ terms we use to discretize the $k(g)$ function, the greater the accuracy we should expect, but for a larger computational cost. Therefore, we have a trade-off to make depending on the application. For climate modelling we require schemes that can accurately compute the radiative forcing of a number of different greenhouse gases over a wide range of concentrations. By contrast, for short-range weather forecasting with present-day greenhouse gas concentrations, the priority is much more on
efficiency: the radiation scheme must be called frequently to capture the local radiative impact of evolving cloud fields, and forecasts must be delivered to customers in a timely fashion. The lower model top in many limited-area weather models also means that, in principle, fewer $k$ terms are required to compute the heating-rate profile. The priorities may be different in other applications of CKD models, such as offline calculations to interpret observations (e.g., Loeb and Kato, 2002), computing the 3D radiative effect of clouds (e.g., Chen and Liou, 2006; Jakub and Mayer, 2016) and providing accurate reference spectra
(e.g., Anderson et al., 1999).

Unfortunately, the tools and know-how to generate new CKD models and to make this accuracy–efficiency trade-off are available to only a handful of specialists worldwide, with the result that most atmospheric models are available with only one gas-optics configuration, which is often not optimized for the application at hand. Indeed, Hogan et al. (2017) surveyed seven models used for the same application of global weather forecasting, and reported that the total number of $k$ terms (shortwave
plus longwave) ranged from 68 to 252.

The purpose of the Correlated K-Distribution Model Intercomparison Project (CKDMIP) is to address these issues. First in CKDMIP we will use benchmark line-by-line calculations to evaluate the accuracy of existing CKD models, followed by the main part of the project in which CKDMIP participants generate new CKD models with different numbers of $k$ terms targeting applications including short-range weather forecasting and climate modelling. Two different band structures are proposed
for them to use. The accuracy versus number of $k$ terms will be computed for each submission, and the results compared to understand how different techniques for constructing CKD models affect their accuracy for the same number of $k$ terms. Finally, it is hoped that the freely available CKDMIP datasets and software will facilitate the development of community tools to allow users to generate their own gas-optics models targeted at specific applications.

The project has similarities to the Radiative Forcing Model Intercomparison Project (RFMIP; Pincus et al., 2016), which
used line-by-line calculations to evaluate the radiation schemes of a number of climate models in terms of surface and top-of-atmosphere (TOA) irradiances for a range of atmospheric profiles and climate scenarios. However, CKDMIP goes further in that it the includes the weather forecasting application, and provides the means to improve the way that CKD schemes make the trade-off between accuracy and efficiency. This is possible by making available the spectral optical depth of each layer of the atmosphere due to each gas separately. The CKDMIP software package allows participants to combine and scale the optical
depths of the gases they are interested in and perform line-by-line radiative transfer calculations on the result, producing their own reference profiles of spectral or broadband irradiances and heating rates.

This protocol paper describes the design and generation of these datasets and software, and what comparisons will be performed. Section 2 describes the overarching design decisions of CKDMIP, including which gases to include, which weather





and climate applications to target, and for climate modelling which range of gas concentrations to consider. Section 3 describes
in detail how the datasets are produced, how the spectral resolution has been chosen and what radiative transfer calculations
are performed. Section 4 then describes what is required of CKDMIP participants, the spectral band structures that should be
used, the metrics that will be used to quantify errors in irradiances and heating rates, and how errors due to the representation of
the spectral variation in cloud properties will be assessed. Section 5 demonstrates the use of the dataset to evaluate an existing,
widely used CKD model.

Finally a note on terminology. Throughout this paper we define a *CKD scheme* as a software component (usually embedded
within the radiation scheme of an atmospheric model) that takes as input profiles of atmospheric temperature, pressure and the
concentrations of a number of gases, and outputs profiles of optical depth for each of a number of $k$ terms. It also includes
a means to compute the Planck function to use for each longwave $k$ term and the TOA solar irradiance for each shortwave $k$
term. A *CKD model* is one configuration of a CKD scheme with a particular number of $k$ terms, which might consist of a set
of look-up tables that can be used by the CKD scheme. A *CKD tool* is a method (which may be fully automated or involve
some hand-tuning) for generating individual CKD models, with some means to control the trade off between accuracy and the
number of $k$ terms.

## 2 Design of evaluation scenarios

### 2.1 Which gases?

The absorption spectra of nine gases are considered in CKDMIP in both the longwave and the shortwave: $H_2O$, $O_3$, $O_2$, $N_2$,
$CO_2$, $CH_4$, $N_2O$, CFC-11 and CFC-12. The first two gases have very variable concentrations and are important in both the
longwave and the shortwave. The concentrations of the second two gases may be treated as fixed both spatially and over the
timescales commonly considered by climate models. $O_2$ is important mainly in the shortwave, but reduces outgoing longwave
radiation (OLR) by around 0.11 W m$^{-2}$ globally (Höpfner et al., 2012). Absorption by $N_2$ is ignored by most operational
radiation schemes, yet it reduces OLR by around 0.17 W m$^{-2}$ (Höpfner et al., 2012), and as will be shown in section 3.6,
has a comparable effect in the shortwave. The concentrations of $N_2$ and $O_2$ are also needed to compute the collision-induced
contribution to the continuum absorption and the broadening efficiency of these molecules, where applicable.

The final five gases listed above are well-mixed greenhouse gases with a significant anthropogenic component. There is a
much larger number of greenhouse gases that could have been included, many of which have a very small individual impact.
However, the purpose of CKDMIP is to evaluate the techniques used by schemes for generating CKD models based on the
different requirements of weather and climate modelling, rather than to produce a single optimum CKD model that explicitly
represents all the greenhouse gases that anyone might want to simulate. Therefore, we have chosen to follow the pragmatic
approach of Meinshausen et al. (2017). They stated that 94.5% of the anthropogenic greenhouse warming (in terms of radiative
forcing) between 1750 and 2014 was due to increases in $CO_2$, $CH_4$, $N_2O$, CFC-11 and CFC-12, with the remaining 5.5%
being attributable to 38 further gases. Their 'Option 2' approximately represents the radiative forcing of these 38 gases by arti-
ficially increasing the concentration of CFC-11 (by around a factor of 3.9 in the present day), and the CMIP6 (Coupled Model



**Table 1.** The three modelling applications of radiation schemes that we envisage would need to be targeted by a different CKD model. The present-day and 'variable' well-mixed greenhouse gas (GHG) concentrations for these scenarios are provided in Table 2. Heating-rate calculations by CKD models will be evaluated at pressures down to the indicated 'lowest pressure', although note that the reference line-by-line calculations are performed down to lower pressures than these.

| Application | Lowest pressure | GHG concentrations |
| --- | --- | --- |
| Limited-area NWP | 4 hPa | Present-day (2020) |
| Global NWP | 0.02 hPa | Present-day (2020) |
| Climate | 0.02 hPa | Variable |

Intercomparison Project Phase 6) historic concentrations and future scenarios are available with these 'CFC-11-equivalent' concentrations. From Cycle 47R1, ECMWF's Integrated Forecasting System will take this approach, using concentrations from the CMIP6 SSP3-7.0 scenario (O'Neill et al., 2016, where 'SSP3-7.0' is the 'regional rivalry' Shared Socioeconomic Pathway of CMIP6, with an anthropogenic radiative forcing of 7.0 W m$^{-2}$ in 2100).

## 2.2 Numerical weather prediction

Table 1 lists the three main applications for which we envisage that CKD models could be optimized. The first two correspond to present-day Numerical Weather Prediction (NWP) at the local and global scale. Both need to represent variable water vapour and ozone, but to a good approximation can assume all other gases to have a constant mole fraction, or to vary as a function of pressure alone. (Note that since the atmosphere is an ideal gas to a good approximation, we can assume the mole fraction of a gas to be equal to its volume mixing ratio.) In principle, this allows the number of $k$ terms to be reduced since, for example, all the well-mixed gases could be merged into a single 'hybrid' or 'composite' gas whose optical properties vary as a function of temperature and pressure alone (e.g., Ritter and Geleyn, 1992; Niemelä et al., 2001).

In terms of the present-day concentrations of the well-mixed gases, we assume that $O_2$ and $N_2$ have constant mole fractions of 0.20946 and 0.78102 mol mol$^{-1}$, respectively, independent of pressure (Jones and Schoonover, 2002). These concentrations are also assumed for all past and future scenarios in section 2.3. The present-day surface concentrations of the five other well-mixed gases are shown in Table 2, and were taken from the CMIP6 SSP3-7.0 scenario for calendar year 2020. The vertical profiles of these gases are discussed in section 3.2.

The difference between the two NWP applications listed in Table 1 is in the location of the model top. The model top quoted for all current configurations of the ECMWF model and all global configurations of the Met Office model used for weather and climate is 0.01 hPa (around 80 km). In the case of the ECMWF model this actually means that the highest model layer spans the pressure range 0–0.02 hPa. Since the temperature of the highest layer of a model is strongly affected by the 'sponge' (Shepherd et al., 1996), we limit evaluation of heating rates to pressures greater than 0.02 hPa. For the limited-area NWP application we evaluate heating rates only for pressures greater than 4 hPa, comparable to the model top used in the Met Office high-resolution UK model.

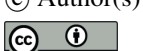



**Table 2.** Surface mole fractions of the five main anthropogenic greenhouse gases for the 34 scenarios considered in CKDMIP, where 'CFC-11 equivalent' is an artificially increased CFC-11 concentration to represent 38 further greenhouse gases (Meinshausen et al., 2017). The present-day scenario will be used to test CKD models developed for the two NWP applications Table 1, while all scenarios will be used to test CKD models for climate. Scenarios 1–18 are used for both the longwave and shortwave evaluation, while scenarios 19–34 (marked with an asterisk) are used for the longwave only. Numbers in bold have been perturbed from their present-day values.

| Scenario | Comment | $CO_2$ ppmv | $CH_4$ ppbv | $N_2O$ ppbv | CFC-11 eq. pptv | CFC-12 pptv |
|---|---|---|---|---|---|---|
| 1 | Glacial maximum | **180** | **350** | **190** | **32** | **0** |
| 2 | Preindustrial | **280** | **700** | **270** | **32** | **0** |
| 3 | Present-day (2020) | 415 | 1921 | 332 | 861 | 495 |
| 4 | Future (2110) | **1120** | **3500** | **405** | **2000** | **200** |
| 5–9 | $CO_2$ forcing | **180, 280, 560, 1120, 2240** | 1921 | 332 | 861 | 495 |
| 10–14 | $CH_2$ forcing | 415 | **350, 700, 1200, 2600, 3500** | 332 | 861 | 495 |
| 15–18 | $N_2O$ forcing | 415 | 1921 | **190, 270, 405, 540** | 861 | 495 |
| *19–20 | CFC-11 forcing | 415 | 1921 | 332 | **0, 2000** | 495 |
| *21–22 | CFC-12 forcing | 415 | 1921 | 332 | 861 | **0, 550** |
| *23–24 | $CO_2/CH_4$ overlap | **180, 2240** | **350** | 332 | 861 | 495 |
| *25–26 | | **180, 2240** | **3500** | 332 | 861 | 495 |
| *27–28 | $CO_2/N_2O$ overlap | **180, 2240** | 1921 | **190** | 861 | 495 |
| *29–30 | | **180, 2240** | 1921 | **540** | 861 | 495 |
| *31–32 | $CH_4/N_2O$ overlap | 415 | **350, 3500** | **190** | 861 | 495 |
| *33–34 | | 415 | **350, 3500** | **540** | 861 | 495 |

## 2.3 Climate modelling

CKD models used for climate modelling should be able to simulate a wide range of greenhouse gas concentrations. The first four lines of Table 2 list individual scenarios that will be tested. They include present-day and preindustrial conditions, plus the conditions at a glacial maximum, with the values for $CO_2$ and $CH_4$ taken from Petit et al. (1999) and for $N_2O$ from the shorter period reported by Schilt et al. (2010). The fourth row shows a 'future' scenario consisting of worst-case conditions for 2110 by extracting the maximum concentrations from any of the CMIP6 scenarios at this time. In this year, the concentration of $CH_4$ peaks at 3500 ppbv in the SSP3-7.0 scenario, and equivalent CFC-11 peaks at 2000 pptv in the SSP5-8.5 scenario.

Scenarios 5–22 in Table 2 show the range of concentrations that will be used in testing the radiative effect of individual gases, keeping all others constant. For each gas we require the capability to simulate the minimum concentrations found in the last million years, which occurred at glacial maxima, up to the maximum concentrations found in any of the CMIP6 future scenarios, which extend until 2250. In the case of $CO_2$ we consider concentrations ranging up to eight times preindustrial. These ranges are very similar to those considered by Etminan et al. (2016). Scenarios 19–22 concern CFC-11 and CFC-12, but





**Table 3.** The four spectral optical depth datasets generated as part of CKDMIP, where $T$ is temperature, $p$ is pressure and $q$ is specific humidity.

| Name | Purpose | Layers | $T$ profiles | Description |
|------|---------|--------|--------------|-------------|
| *Evaluation-1* | Training & evaluation | 54 | 50 | Realistic profiles selected from NWP-SAF dataset |
| *Evaluation-2* | Independent evaluation | 54 | 50 | Further profiles selected from NWP-SAF dataset |
| *MMM* | Training | 52 | 3 | Median, min. and max. of NWP-SAF $T$, $q$ and $O_3$ profiles |
| *Idealized* | Generating look-up tables | 53 | 11 | Idealized profiles regularly spaced in $T$, $\log p$ and $\log q$ |

as will be shown from line-by-line calculations in section 3.6, the magnitude of their instantaneous TOA and surface shortwave radiative forcing is less than 0.002 W m$^{-2}$, so these scenarios are used only for longwave evaluation.

Etminan et al. (2016) reported that due to the overlap of the absorption spectra of $CO_2$, $CH_4$ and $N_2O$, the longwave radiative forcing associated with changing the concentration of one of these gases can depend on the concentration of the other two. To test the ability of CKD models to simulate this effect, the final 12 scenarios in Table 2 perturb the concentrations of pairs of these gases to their extreme values, while keeping the others at present-day concentrations. These scenarios are also only for longwave evaluation since we calculate that overlap effects change shortwave TOA forcings by only of order 0.001 W m$^{-2}$.

In principle, there are important applications in addition to those shown in Table 1, such as atmospheric reanalysis, which have been generated back to the mid-19th century (e.g. Compo et al., 2011). A CKD model targeted at this application would only need to span greenhouse gas concentrations from preindustrial to present-day. We decided not to include this application in CKDMIP, partly not to overload the participants, but also because of the expectation that the number of $k$ terms required would not be very different between the reanalysis and climate modelling applications.

## 3    Generating datasets


Table 3 lists the four CKDMIP datasets. Each consists of profiles of layer-wise spectral optical depth due to individual gases. The first two (*Evaluation-1* and *Evaluation-2*) each consist of 50 realistic profiles of temperature, water vapour and ozone (described in section 3.1), accompanied by vertical profiles of the well-mixed gases (described in section 3.2). *Evaluation-1* is provided to participants and may be used to train individual CKD models, while *Evaluation-2* is held back to provide

independent evaluation. Section 3.3 describes the last two datasets, which could also be useful in the training of new CKD models. Section 3.4 then describes how the profiles of spectral optical depth were computed for each dataset. Section 3.5 describes the radiative transfer calculations performed on these absorption spectra, an example of which is given in section 3.6 where we estimate the longwave and shortwave radiative importance of each of the seven well-mixed gases.

### 3.1    Temperature, humidity and ozone

For evaluating radiation schemes in RFMIP, Pincus et al. (2016) extracted a set of 100 contrasting atmospheric profiles from the 60-layer ERA-Interim reanalysis dataset, whose highest model level spans the pressure range 0–0.2 hPa. As well as being





ten times greater than the pressure of the highest model level in the current ECMWF and Met Office global models, this vertical grid not sufficient to fully resolve the strong peak in atmospheric heating and cooling rates that occurs at the stratopause, nor to test solar absorption by carbon dioxide in the mesosphere.

Therefore, we have selected a new set of temperature, pressure, humidity and ozone profiles from the 25,000 'NWP-SAF' profiles of Eresmaa and McNally (2014), which they extracted from ECMWF operational model forecasts in 2013 and 2014. By this time the model used 137 layers with the highest layer spanning pressures 0–0.02 hPa, as in its current configuration. As in the ECMWF operational model, CKDMIP assumes a hydrostatic atmosphere, in which case the mass of a layer is defined purely from the pressure at the layer interfaces and the acceleration due to gravity.

The 50 profiles of the '*Evaluation-1*' dataset consist of 33 randomly taken from the NWP-SAF dataset. An additional 17 profiles are selected to contain the extreme values (both maximum and minimum) in the entire dataset of (a) temperature in the layer nearest the surface, (b) temperature at 400 hPa, (c) temperature at 100 hPa, (d) temperature at 10 hPa, (e) temperature at 1 hPa, (f) specific humidity at 400 hPa, (g) specific humidity at 100-hPa (maximum only), (h) ozone concentration at 10-hPa, and (i) ozone concentration at 1 hPa.

The '*Evaluation-2*' dataset, intended to provide independent evaluation of the CKD models, uses a different set of 33 random profiles from the NWP-SAF dataset, along with 17 profiles containing the extreme values at different levels from those used by *Evaluation-1*, specifically (a) temperature where the pressure falls to 90% of its surface value, (b) temperature at 200 hPa, (c) temperature at 50 hPa, (d) temperature at 5 hPa, (e) temperature at 0.5 hPa, (f) specific humidity at 200 hPa, (g) specific humidity at 50 hPa (maximum only), (h) ozone concentration at 5 hPa and (i) ozone concentration at 0.5 hPa.

It was apparent from inspection of the data that there was virtually no variability in stratospheric water vapour in the ECMWF model at the time the NWP-SAF profiles were generated, which is a problem for training and evaluating a gas-optics model. Therefore, additional variability has been added by multiplying the humidity profiles by the following function of pressure, $p$:

$$f(p,r) = \exp\left[ r \times \frac{1 - \mathrm{erf}\left( \frac{p - 100 \text{ hPa}}{50 \text{ hPa}} \right)}{2} \right], \tag{1}$$

where $r$ is a random number drawn from a Normal distribution with mean of zero and standard deviation 0.25, and is constant
for each individual profile. This function adds around 25% variability in the stratosphere and mesosphere, but leaves the troposphere virtually unchanged. Unrealistically low humidities have been removed by setting the minimum specific humidity to $10^{-7}$ kg kg$^{-1}$. The resulting temperature, humidity and ozone mixing ratios are shown by the red and blue lines in Fig. 1.

Training and evaluating a CKD model is costly both in terms of computation and storage due to the high spectral resolution required, and 137 layers is more than needed for evaluating clear-sky radiative transfer. Therefore, we interpolate the profiles on
to a coarser grid with 54 layers. We use the Line-By-Line Radiative Transfer Model (LBLRTM; Clough et al., 2005), version 12.8, which takes as input the temperature, pressure and gas concentrations at the interfaces between layers. The highest two layers of the coarser grid are bounded by pressures of 0.0001, 0.01 and 0.02 hPa; the first of these represents the TOA since LBLRTM cannot compute gas properties at zero pressure. As shown in Table 1, the pressure surfaces 0.02 and 4 hPa mark the point at which evaluation of heating rates begins. We assign 15 layers between these two pressure surfaces, with the interfaces



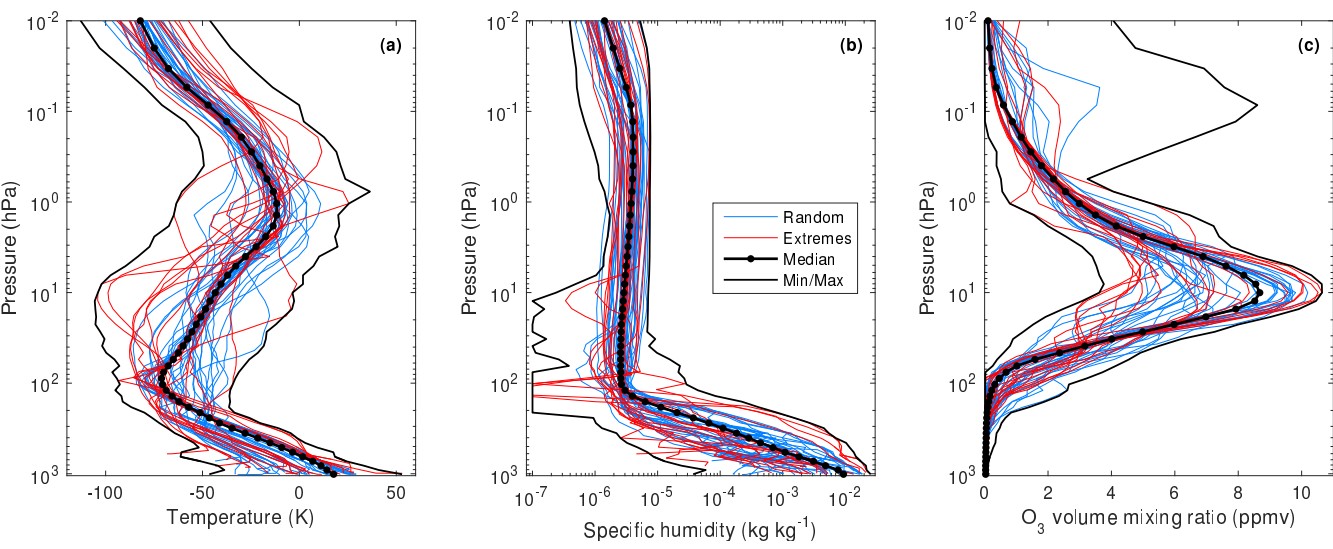

**Figure 1.** Vertical profiles of the temperature, specific humidity and ozone concentration for the '*Evaluation-1*' dataset described in section 3.1.

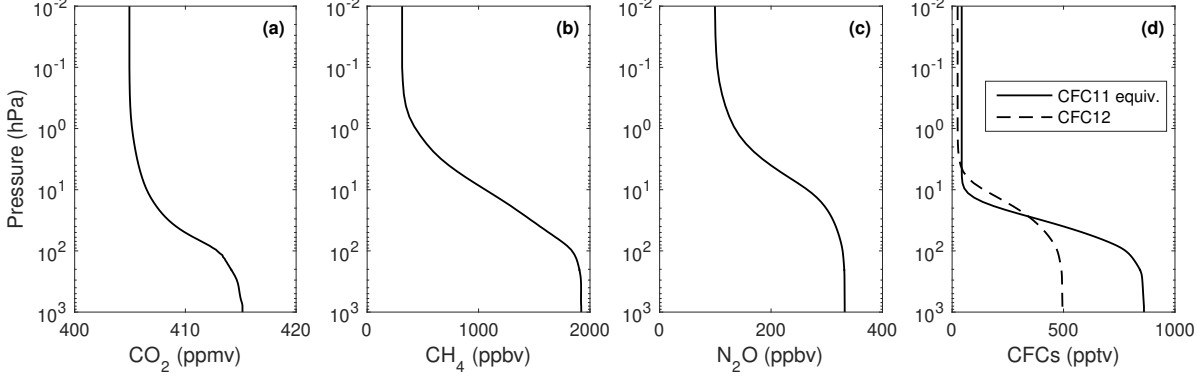

**Figure 2.** Vertical profiles of the five well-mixed greenhouse gases for the present-day (2020) surface concentrations listed in Table 2.

between them spaced linearly in $p^{0.15}$ space. The pressures defining the remaining layers vary according to the surface pressure $p_s$: we assign 35 layers between 4 hPa and $p_s/1.005$, again spaced linearly in $p^{0.15}$ space. Finally, a further two layers are added very close to the surface (bounded by $p_s/1.005$, $p_s/1.002$ and $p_s$) in order to resolve sharp temperature gradients in the surface layer. The black dots in Fig. 1 mark the corresponding interfaces between layers for the median profiles described in section 3.3.



## 3.2 Well-mixed gases

Many weather and climate models assume a spatially constant mole fraction for each of the well-mixed gases, whereas for a little more realism they should decrease with height. The radiation scheme in the ECMWF model uses climatologies of these gases that vary with month, latitude and pressure, with the $CO_2$ and $CH_4$ climatologies taken from the MACC analysis system (Inness et al., 2013) and the $N_2O$, CFC-11 and CFC-12 climatologies from the Cariolle chemistry model (Bechtold et al., 2009). Long-term changes due to anthropogenic emissions are represented by scaling these fields so that the global-mean surface values match either historic measurements (for hindcasts and reanalysis) or the CMIP6 SSP3-7.0 scenario (for operational forecasts from model cycle 47R1). We have averaged these climatologies globally and annually, and scaled them to the 2020 surface values in SSP3-7.0, to obtain the profiles shown in Fig. 2. Present-day $CO_2$ has a difference of 10 ppmv between the values at 1000 and 0.01 hPa. In the case of CFC-11 and CFC-12, the concentrations from the Cariolle model drop to almost zero in the upper stratosphere and mesosphere, which could be problematic for using them to train the pressure dependence in a CKD model. Therefore, the profiles of these two gases have been artificially modified to fall to no less than 5% of their surface value. In order to obtain profiles with the surface concentrations shown in Table 2, we simply scale the profiles shown in Fig. 2.

We have computed that the difference in the TOA longwave radiative forcing of a gas with a constant mole fraction with pressure, versus the more realistic profiles in Fig. 2 but the same surface concentration, is 10% for CFC-11, 5% for CFC-12, and less than 0.2% for the other three gases.

## 3.3 Additional training datasets

Two additional datasets are shown at the bottom of Table 3, which are intended to facilitate the development of CKD schemes, while being consistent with the datasets that will be used to evaluate them. The '*MMM*' dataset contains the optical properties of all nine gases but using the median, minimum and maximum temperature profiles derived from the entire 25,000-profile NWP-SAF dataset; these temperatures are shown by the black lines in Fig. 1. In the case of $H_2O$ and $O_3$ only, three concentration profiles are used for each temperature, corresponding also to the median, minimum and maximum of the NWP-SAF profiles (shown in Figs. 1b and 1c). For all other gases the present-day concentrations shown in Fig. 2 are used. The vertical grid is the same as for the *Evaluation-1* and *Evaluation-2* datasets, except that surface pressure is set to mean sea level pressure ($p_s = 1013.25$ hPa), and the two layers very close to the surface are not used so that the total number of layers is 52 rather than 54.

The final '*Idealized*' dataset contains absorption spectra for idealized temperature and concentration profiles that are intended to cover the full range of likely temperature, pressure and concentrations found in the atmospheres that any CKD model would be applied to. Therefore, they can be used to populate look-up tables of molar absorption to be used by CKD models. We envisage that the maximum layer-mean pressure that needs to be accommodated by a radiation scheme is 1100 hPa, so construct a logarithmically spaced pressure profile of 53 elements, containing ten points per decade with a maximum layer-mean pressure of 1100 hPa. At each pressure, 6 temperatures are simulated spanning a 100 K range at 20 K intervals. We



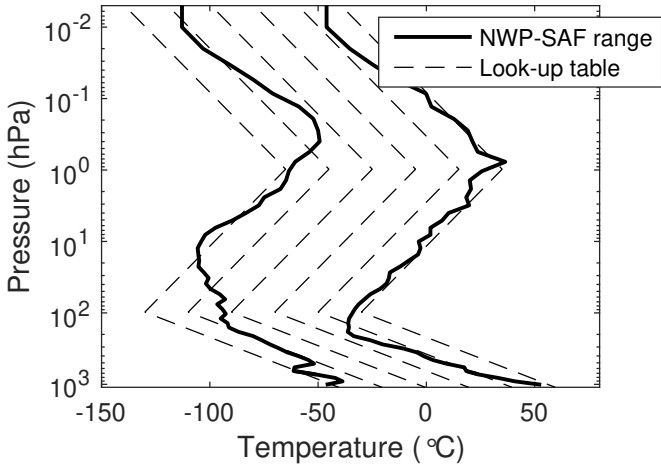

**Figure 3.** The solid lines show the minimum and maximum temperatures of the NWP-SAF dataset, also shown in Fig. 1a. The dashed lines show the 6 idealized temperature profiles, 20 K apart, used in the '*Idealized*' dataset in Table 3.

use idealized temperature profiles shown in Fig. 3 that are intended to encompass the maximum and minimum temperatures found in the NWP-SAF dataset. For all gases, absorption spectra are computed for mole-fraction profiles that are constant with

pressure, using the present-day values for the five well-mixed gases shown in Table 2, and 5 ppmv for $O_3$. Since the molar absorption of these gases is very close to constant with concentration (see section 3.4), only one concentration needs to be simulated for each. In the case of water vapour whose absorption varies with concentration, we simulate 12 logarithmically spaced specific humidities from $10^{-7}$ to $10^{-1.5}$ kg kg$^{-1}$, i.e., using two values per decade.

### 3.4    Line-by-line modelling

The spectral optical depths of the individual gases have been computed using version 12.8 of the Line-by-Line Radiative Transfer Model (LBLRTM) (Clough et al., 2005) developed at Atmospheric & Environmental Research (AER). LBLRTM incorporates the self- and foreign-broadened water vapour continuum via the Mlawer-Tobin-Clough-Kneizys-Davies (MT_CKD) continuum model, version 3.2 (Mlawer et al., 2012). Continua for $CO_2$ and for the collision induced bands of $O_2$ and $N_2$ are also included in the computations. Line coupling for $CO_2$ is treated as first order with coefficients computed as specified by

Lamouroux et al. (2015). It should be noted, however, that line coupling coefficients for the 30012←00001 and 30013←00001 bands of the main isotopologue (at 6348 cm$^{-1}$ and 6228 cm$^{-1}$, respectively) have been calculated from the tridiagonal relaxation matrix parameters of Devi et al. (2007a, b). The spectroscopic input parameters have been taken from the AER line parameter database, version 3.6, which is largely drawn from HITRAN 2012 (Rothman et al., 2013) but with AER customized modifications, most notably for $H_2O$, $CO_2$ and $O_2$. The AER line parameters for $CH_4$ include line coupling parameters for the

$\nu_3$ (3000 cm$^{-1}$) and $\nu_4$ (1300 cm$^{-1}$) bands of the main isotopologue.



**Figure 4.** (a) The normalized cumulative distribution function (CDF) of the Planck function at four temperatures spanning those found in the atmosphere and the surface in Fig. 1a; (b) the CDF of the solar spectral irradiance (integrating to 1361 W m$^{-2}$); and (c) the absorption spectra of the nine gases considered in CKDMIP at 100 hPa from the 'median' profile of the *MMM* dataset with present-day concentrations of the well-mixed gases (except for CFC-11 which is at a higher 'equivalent' concentration). The red vertical lines in panels a and b indicate the boundaries of the bands defined in section 4.1 for the longwave and shortwave calculations, respectively, while the grey shading indicates parts of the spectrum not considered in the longwave and shortwave calculations. The vertical black dotted line in each panel at a wavenumber of 2500 cm$^{-1}$ (wavelength of 4 $\mu$m), indicates where the horizontal scale changes from linear in wavenumber to the left of the line, to logarithmic in wavenumber to the right.





Rather than defining radiation as 'longwave' or 'shortwave' depending on whether its wavenumber is less than or greater than some specific value, we define the longwave as any radiation originating from emission by the surface or atmosphere, and shortwave as any radiation originating from the sun. The longwave spectrum is taken to span the wavenumber range 0–3260 cm$^{-1}$, which covers 99.997% of the Planck function at 0°C and 99.971% at +50°C. The shortwave spectrum is taken to span the range 250–50,000 cm$^{-1}$, which misses only 0.012 W m$^{-2}$ of the solar irradiance at wavenumbers less than 250 cm$^{-1}$ and 0.103 W m$^{-2}$ at wavenumbers greater than 50,000 cm$^{-1}$. These ranges are shown in the top two panels of Fig. 4, while the bottom panel shows the spectral absorption of the nine gases at 100 hPa for the 'median' profile of the *MMM* dataset using present-day concentrations of the well-mixed gases.

An important practical consideration is to determine at what spectral resolution to produce the absorption spectra. They need to be fine enough resolution that the most narrow spectral lines are resolved and the resulting irradiance and heating-rates profiles are an accurate benchmark, but also a manageable data volume for storage, processing and distribution. LBLRTM can inform the user of the spectral resolution it needs to resolve the lines at a particular pressure, and for $CO_2$ in the longwave at 0.01 hPa (the most important gas at the pressure where the lines are finest), it recommends a wavenumber resolution such that more than 20 million spectral points are required. Using this resolution as a reference, we have experimented with degrading the spectral resolution in four spectral ranges bounded by the wavenumbers 0, 350, 1300, 1700 and 3260 cm$^{-1}$. Computing the heating rate error for each spectral range leads us to adopt spectral resolutions of 0.0002, 0.001 and 0.005 cm$^{-1}$ in the three spectral ranges 0–1300, 1300–1700 and 1700–3260 cm$^{-1}$, respectively. This leads to heating-rate errors of no more than around 0.005 K d$^{-1}$ (all of which occur in the upper stratosphere and mesosphere) in any of the four original wavenumber ranges, even for the most challenging scenario of 8 times preindustrial concentrations of $CO_2$. This leads to 7,211,999 spectral points in the longwave.

A similar approach has been taken in the shortwave, resulting in spectral resolutions of 0.002, 0.001, 0.002, 0.02 and 1 cm$^{-1}$ in the spectral ranges 250–2200, 2200–2400, 2400–5150, 5150–16000 and 16000–50000 cm$^{-1}$, respectively. For overhead sun this also leads to heating-rate errors of no more than around 0.005 K d$^{-1}$ in any of these wavenumber ranges, for 8 times preindustrial $CO_2$. This leads to 3,126,494 spectral points in the shortwave.

A further significant reduction in data volume is possible if the absorption cross-section per molecule is independent of the concentration of that gas, so varies only as a function of temperature and pressure. In this case, for well-mixed gases, the profile of layer-wise optical depth need only be provided for a single concentration profile; if optical depths are required for concentration profiles scaled by a constant, then the optical depths themselves may simply be scaled. We have computed absorption spectra for each gas over the full range of concentrations required in Table 2, and found that to a very good approximation molar absorption can be treated as independent of concentration for all gases except water vapour. Therefore, for the well-mixed gases, absorption spectra are provided only for present-day concentrations. The CKDMIP software accordingly allows the user to scale the optical depth of each gas before performing radiative transfer calculations on the mixture.

The CKDMIP software calculates the spectral optical depth due to Rayleigh scattering using the model of Bucholtz (1995), in which the per-molecule Rayleigh scattering cross section, in m$^2$, is given by the following for wavelengths of less than





275  0.5 $\mu$m:

$$\sigma_r = 3.01577 \times 10^{-32} \lambda^{3.55212+1.35579\lambda+0.11563/\lambda}, \tag{2}$$

where wavelength $\lambda$ is in $\mu$m, and by the following for wavelengths greater than 0.5 $\mu$m:

$$\sigma_r = 4.01061 \times 10^{-32} \lambda^{3.99668+0.00110298\lambda+0.0271393/\lambda}. \tag{3}$$

A realistic TOA solar irradiance spectrum was extracted from the climate data record of Coddington et al. (2016) by averag-
280  ing over the last 33 years (1986–2018 inclusive), i.e., three solar cycles. It has a resolution of 1 nm at wavelengths shorter than
750 nm, and is interpolated to the spectral resolution of the shortwave gas absorption spectra.

As stated above, the water vapour spectra include the continuum computed using the MT_CKD model, but there is still
considerable uncertainty on the strength of the water vapour continuum, particularly in the near infrared (Shine et al., 2016),
and indeed this could be a source of difference between individual gas optics schemes and the reference calculations produced
in CKDMIP. Therefore, for each dataset, we produce an additional set of water vapour files but with no representation of the
continuum. If needed, evaluation can be carried out using only the contribution from spectral lines, or alternatively different
models of the continuum can be tried.

The absorption spectra are stored, one gas per file, in netCDF4/HDF5 format with compression, so the file size depends on
the spectral extent and degree of fine structure in the spectrum. In the longwave, the volume of a single file (containing 10
profiles) varies from 0.5 GB for CFC-11 to around 10 GB for $CH_4$, and the 50-profile *Evaluation-1* dataset amounts to 222 GB
in total. In the shortwave the *Evaluation-1* dataset amounts to 109 GB.

### 3.5  Generating irradiance and heating-rate benchmarks

The CKDMIP software takes as input the spectral optical depths of each of a number of gases, optionally scales the opti-
cal depths of the well-mixed gases if a different concentration is required, and computes clear-sky aerosol-free irradiances
(broadband or spectral) at layer interfaces for each of the test profiles. These can be used to compute broadband or spectral
heating-rate profiles. The intention is that the radiative transfer equations are then the same as those used by large-scale at-
mospheric models, and the same solver is used with the various CKD models in order that any differences to the line-by-line
broadband irradiances are due to the representation of gas optics, not the details of the solver.

In the longwave we use a no-scattering solver with the following properties:

– Surface emissivity is assumed to be unity.

– The skin temperature of the surface is assumed to be equal to the air temperature at the base of the lowest atmospheric
layer.

– Local thermodynamic equilibrium is assumed.





**Table 4.** Instantaneous radiative forcing at top-of-atmosphere (TOA) and the surface of each of the seven well-mixed gases at present-day concentrations, compared to setting their concentration to zero while leaving the other gases unchanged. The values are from line-by-line radiative transfer calculations using the settings in section 3.5, averaging over the 50 *Evaluation-1* profiles. The shortwave calculations are averaged over the five zenith angles so represent a daytime average. Since there is substantial profile-to-profile variation, only two significant figures are shown.

|  | $N_2$ | $O_2$ | $CO_2$ | $CH_4$ | $N_2O$ | CFC-11 eq. | CFC-12 |
|---|---|---|---|---|---|---|---|
| Longwave TOA | 0.15 | 0.096 | 22 | 1.5 | 1.5 | 0.22 | 0.17 |
| Longwave surface | 0.067 | 0.0069 | 21 | 0.86 | 0.91 | 0.23 | 0.17 |
| Shortwave TOA | 0.062 | 1.25 | 0.57 | 0.32 | 0.054 | 0.00042 | 0.00053 |
| Shortwave surface | $-0.24$ | $-4.3$ | $-2.7$ | $-1.2$ | $-0.27$ | $-0.0014$ | $-0.0017$ |

– The angular distribution of radiation is approximated by four discrete zenith angles in each hemisphere (8 streams in total), chosen using the rules of Gauss-Legendre Quadrature. The software supports between one and eight angles, although we find that broadband longwave calculations with four angles agree with those from eight to within 0.05 W m$^{-2}$ in terms of irradiances and 0.02 K d$^{-1}$ in terms of heating rates.

– The temperature at layer interfaces is taken as input and a linear-in-optical-depth variation of the Planck function within each layer is assumed, leading to the use of Eqs. 6–12 of Hogan and Bozzo (2018).

The shortwave scheme has the following characteristics:

– The surface is assumed to be a Lambertian reflector with an albedo of 0.15, the global mean value according to Wild et al. (2013).

– It uses a direct-beam calculation plus a two-stream diffuse calculation, with the Zdunkowski et al. (1980) coefficients characterizing the rate of exchange of energy between the three streams, and the Meador and Weaver (1980) solutions to the two-stream equations in individual layers. While two streams is fewer than used in the longwave, it is of sufficient accuracy because shortwave gaseous absorption in clear skies is predominantly by the direct solar beam.

– Calculations are performed at five values of the cosine of the solar zenith angle ($\mu_0$): 0.1, 0.3, 0.5, 0.7 and 0.9. This even sampling is appropriate given that the sunlight striking the Earth during daytime has a uniform $\mu_0$ distribution between 0 and 1. We do not account for the fact that individual test profiles at a particular latitude would each experience a different $\mu_0$ distribution.

– No account is made for Earth curvature.

The atmospheric heating rate in layer $i$ is computed from the net irradiance divergence across a layer, as:

$$\frac{dT_i}{dt} = -\frac{g_0}{C_p}\frac{F^n_{i+1/2} - F^n_{i-1/2}}{p_{i+1/2} - p_{i-1/2}}, \tag{4}$$





where $p_{i+1/2}$ and $F_{i+1/2}^n$ are the pressure and net downward irradiance, respectively, at the interface between layers $i$ and $i+1$

(counting down from TOA), $g_0$ is the acceleration due to gravity (standard gravity) and $C_p$ is the specific heat of dry air, taken to be constant at $1004 \text{ J kg}^{-1} \text{ K}^{-1}$.

### 3.6 Radiative forcing of well-mixed gases

Many current CKD models omit some of the gases considered in CKDMIP, particularly in the shortwave. Table 4 provides an estimate of the instantaneous radiative forcing of individual well-mixed gases at present-day concentrations, compared to

setting their concentration to zero, computed from averaging over line-by-line calculations on the 50 *Evaluation-1* profiles. This is not an accurate estimate of the climatic impact of each gas since it neglects clouds and fast stratospheric adjustment, and the profiles are not necessarily globally representative, but it gives an indication of the error incurred by neglecting particular gases. The longwave impacts of $N_2$ and $O_2$, ignored by many CKD models, are similar to the values reported by Höpfner et al. (2012). Most shortwave CKD models ignore $N_2$ and $N_2O$, but the results here indicate that this leads to an overestimate of

daytime clear-sky net surface solar irradiance by around $0.5 \text{ W m}^{-2}$. It would be interesting to investigate the impact of this on the climate of a global model.

## 4 CKDMIP experimental protocol

Anyone with a CKD tool can take part in CKDMIP. Participants are provided with access to the *Evaluation-1*, *MMM* and *Idealized* datasets, and the software described in section 3.5 to perform line-by-line radiation calculations on them. They may

use these or their own datasets as input to their CKD tool. In section 4.1 we describe the band structure that should be used by participants, if possible. Section 4.2 describes the calculations that should be performed by participants and the data they provide. In section 4.3 we outline the how these data are processed to quantify accuracy, and to investigate the accuracy–efficiency trade-off.

### 4.1 Common band structures

Virtually all operational CKD models for weather and climate split the longwave and shortwave spectra into bands, and compute $k$ distributions within each one. As shown in the survey of Hogan et al. (2017), the number of bands is strongly correlated to the total number of $k$ terms, and therefore to the overall computational efficiency of a CKD model. The choice of bands can be dependent on the constraints of a particular CKD scheme: some require the longwave bands to be narrow enough that the Planck function may be assumed constant (e.g. Fu and Liou, 1992); some need to restrict the number of active gases in a band

(e.g. Mlawer et al., 1997); some assume the spectral overlap of different gases is random, invalid for wide bands (e.g. Ritter and Geleyn, 1992); while most assume that cloud and surface properties are constant within each band, which could lead to significant errors in the shortwave if the bands are too wide (Lu et al., 2011). All of these arguments deserve detailed scrutiny within CKDMIP.





**Table 5.** The spectral boundaries of the (left) 'narrow' and (right) 'wide' longwave bands, in which participants are asked to generate CKD models. The narrow bands are essentially the same as those in RRTMG, except for the final band which spans the last four bands of RRTMG. The band boundaries are depicted in Fig. 4a.

| | Narrow bands | | | Wide bands |
|---|---|---|---|---|
| # | Spectral interval ($cm^{-1}$) | RRTMG $k$ terms | # | Label |
| 1 | 0–350 | 8 | 1 | Far infrared |
| 2 | 350–500 | 14 | | |
| 3 | 500–630 | 16 | 2 | Main $CO_2$ band |
| 4 | 630–700 | 14 | | |
| 5 | 700–820 | 16 | | |
| 6 | 820–980 | 8 | 3 | Infrared window |
| 7 | 980–1080 | 12 | | |
| 8 | 1080–1180 | 8 | | |
| 9 | 1180–1390 | 12 | 4 | Mid-infrared A |
| 10 | 1390–1480 | 6 | | |
| 11 | 1480–1800 | 8 | | |
| 12 | 1800–2080 | 8 | 5 | Mid-infrared B |
| 13 | 2080–3260 | 10 | | |

We propose two band structures, shown in Table 5 for the longwave and Table 6 for the shortwave. Since RRTMG (Mlawer

et al., 1997) is so widely used, our proposed 'narrow bands' are modelled on RRTMG, except that we merge a few of the very narrow or very low-energy bands that RRTMG represented with four or fewer $k$ terms. This leads to 13 bands in both the longwave and shortwave. These bands should be narrow enough to satisfy all the needs for narrowness cited previously. To assist participants who do not wish to download all the large spectral absorption files, much smaller files are available containing benchmark irradiance profiles computed for each scenario of the *Evaluation-1* dataset, both broadband values and

values averaged in each of the narrow bands.

The 'wide bands', of which there are five in both the longwave and the shortwave, consist of groupings of the narrow bands. In the longwave these are purposefully somewhat wider than in most current CKD models, in order to really test the limits of some of the restrictions cited above. The wide-band models will be compared to the narrow-band models in terms of both accuracy and efficiency, which may allow the advantages of CKD schemes that do not assume the Planck function to be

constant across a band, or do not assume random spectral overlap, to become apparent.





**Table 6.** As Table 5 but for the shortwave. The narrow bands are as in RRTMG, except for band 7 which spans two RRTMG bands. The band boundaries are depicted in Fig. 4b.

| | *Narrow bands* | | | *Wide bands* |
| | Spectral | RRTMG | | |
| # | interval (cm$^{-1}$) | $k$ terms | # | Label |
|---|---|---|---|---|
| 1 | 250–2600 | 12 | | |
| 2 | 2600–3250 | 6 | 1 | Mid-infrared |
| 3 | 3250–4000 | 12 | | |
| 4 | 4000–4650 | 8 | | |
| 5 | 4650–5150 | 8 | 2 | Shortwave infrared |
| 6 | 5150–6150 | 10 | | |
| 7 | 6150–8050 | 12 | | |
| 8 | 8050–12850 | 10 | 3 | Near infrared |
| 9 | 12850–16000 | 8 | | |
| 10 | 16000–22650 | 6 | 4 | Visible window |
| 11 | 22650–29000 | 6 | | |
| 12 | 29000–38000 | 8 | 5 | Ultraviolet |
| 13 | 38000–50000 | 6 | | |

Some participants may wish to use their own sub-bands within these wide bands if they think it will achieve a better accuracy–efficiency trade-off for a particular wide band. For example, Cusack et al. (1999) used two 'split bands' in the longwave, one which represented the wings of the main $CO_2$ band (essentially a merger of our narrow bands 3 and 5) and the other which represented the parts of the infrared window on either side of the ozone band (essentially a merger of our narrow bands 6 and 8).

Finally, CKDMIP welcomes submissions using even wider bands. Indeed, the 'full-spectrum correlated-$k$' (FSCK) technique has been proposed as a means to achieve good accuracy using only one band in the longwave (Hogan, 2010) and two in the shortwave (Pawlak et al., 2004). The investigation of the effect of spectral variations of cloud properties within bands and $k$ terms described in section 4.4 will be particularly important for FSCK submissions.

All submissions, whether using 'narrow', 'wide' or other band structures, will be compared to each other according to their broadband accuracy and their overall efficiency (total number of $k$ terms).

## 4.2 Contribution of CKDMIP participants

Ideally, CKDMIP participants would use their tool to generate a CKD model for all combinations of the following:





– The longwave and shortwave.

– The three applications listed in Table 1.

– The narrow and wide band structures described in section 4.1 (and optionally even wider bands).

– A range of total number of $k$ terms (at least three configurations), in order that the efficiency–accuracy trade-off can be explored.

This could potentially lead to a full submission involving the generation of 36 CKD models. It is recognized that this is
potentially very demanding, so reduced submissions are welcome according to the scientific interests of the participant. In principle, a participant could submit just one longwave and one shortwave CKD model; if it used the narrow bands specified in section 4.1 and targeted climate modelling, then it could still be tested against other models in all scenarios.

Participants do not submit the code for their CKD models, but rather run each of them on the 50-profile *Evaluation-1* dataset. For CKD models generated for the two NWP applications, the well-mixed greenhouse gas concentrations use the present-day
values given in the third line of Table 2. For CKD models generated for climate modelling, they run each of the 34 scenarios given in Table 2 in the longwave, and the first 18 scenarios in the shortwave.

For each of these scenarios, they submit a netCDF file containing the following variables as a function of profile number in the *Evaluation-1* dataset:

– Pressure at layer interfaces, copied from the input file;

– The absorption optical depth of all gases in each layer, in each of $N$ $k$-terms;

– In the shortwave only, the Rayleigh scattering optical depth in each layer and $k$ term;

– In the shortwave only, the TOA solar irradiance integrated over the parts of the spectrum contributing to each $k$ term, scaled such that these numbers sum to a total solar irradiance of 1361 W m$^{-2}$.

– In the longwave only, the Planck function at each layer interface, integrated over the parts of the spectrum contributing
to each $k$ term. At a given layer interface, these values should sum to $\sigma T^4$, where $\sigma$ is the Stefan-Boltzmann constant and $T$ is the temperature at the layer interface (provided in the input file).

These files should be compatible with the CKDMIP software, which can then read them in and compute profiles of upwelling and downwelling irradiances, both at each $k$ term and as broadband values. This ensures that the radiative transfer is identical to that used in generating the line-by-line benchmarks, so that when the irradiances are compared to the benchmarks, the
differences are only due to the spectral approximations made in the CKD model.

A further file is required for each CKD model generated, describing which parts of the spectrum are represented by each $k$ term, to be used in section 4.4 for investigating the representation of cloud optical properties. In the longwave this should be expressed at a resolution of 10 cm$^{-1}$ and in the shortwave at a resolution of 50 cm$^{-1}$. This is commensurate with the spectral scale at which the optical properties of clouds vary.



After the first phase of comparisons using the *Evaluation-1* dataset in which the line-by-line benchmarks are made available to participants, a second phase of comparisons will be conducted using *Evaluation-2* dataset, in which the line-by-line benchmarks are withheld.

The protocol above assumes that participating radiation schemes have a clean separation between the generation of optical depths in each $k$ term and the radiative transfer performed on them. Allowance will need to be made for some schemes in

which the separation is not so clean. For example, SOCRATES (the Suite Of Community Radiation codes based on Edwards and Slingo, 1996) uses the concept of 'equivalent extinction' to treat minor gases (Edwards, 1996). In the longwave this involves performing $M$ no-scattering radiation calculations to work out the contribution from minor gases in a band. The net irradiance from these profiles are analyzed to work out the equivalent extinction, which is then added to the $N$ $k$-terms for representing the major gases in the band. A full longwave radiative transfer calculation, including scattering, is then performed on these $N$ $k$-

terms. This approach could be accommodated in CKDMIP by the participant performing the $M$ initial calculations themselves and providing the resulting $N$ optical depth profiles. The CKDMIP radiative transfer software would then be run on these $N$ $k$-terms (verifying that it gives very similar results to the SOCRATES radiative transfer solver), but when assessing the accuracy–efficiency trade-off, the cost of the scheme would be counted as $aM + N$, the $a$ factor being optionally less than one to account for the fact that equivalent extinction can be computed with a cheaper solver.

In the shortwave, the SOCRATES scheme uses a more sophisticated treatment of gas optics ($M + N$ $k$-terms) for the cheap direct-beam radiative transfer calculation, and a simpler treatment of gas optics ($N$ $k$-terms) for the more expensive solver for scattered radiation. This could be accommodated by the participant providing CKDMIP with separate direct and a diffuse optical depths in the $N$ $k$-terms, and again the cost of the scheme being counted as $aM + N$, with $a$ this time representing the cost of the direct-only versus full shortwave radiation calculation.

## 4.3   Error metrics

The irradiance profiles computed from the submissions of participants for the relevant scenarios in Table 2 will be compared to the equivalent line-by-line benchmarks, with differences in upwelling and downwelling irradiances being characterized by the bias and root-mean-squared error (RMSE) over the 50 profiles. Particular emphasis will be placed on the surface downwelling and TOA upwelling irradiances.

Atmospheric heating-rate bias and RMSE will be examined as a function of pressure. The profile of heating-rate error will be summarized by a few error metrics, such as the whole-profile RMSE, or the values for the troposphere, stratosphere and (except for the 'limited-area NWP' application) mesosphere separately. An appropriate weighting with pressure will need to be specified; rather than weighting linearly with pressure, which overweights the troposphere, Hogan (2010) proposed weighting by the square-root of pressure, which increases the weighting of stratospheric errors, but other powers (e.g., the cube-root) are

possible. Naturally, the heating-rate errors will only be counted down to the lowest pressure for the application in question (see Table 1). The handful of RMSE values will then be plotted as a function of number of $k$ terms to compare how different CKD tools perform in terms of accuracy versus efficiency.





In addition, we will look at the accuracy of the CKD models for climate in terms of the TOA and surface radiative forcing they predict when the five well-mixed anthropogenic greenhouse gases are perturbed as described by the scenarios in Table 2.
This will involve simple averaging over the 50 profiles.

Note that we do not propose to define a 'cost function' that combines multiple error measures into a single metric, as it may not align with those used explicitly or implicitly by the various CKD tools. Nonetheless, all model output will be freely available for participants to compute their own error metrics should they wish.

### 4.4  Errors due to the spectral variation of cloud properties

Until this point, we have considered exclusively clear-sky radiation calculations with a spectrally constant surface albedo. It is known that errors can arise in cloudy skies if cloud optical properties are assumed constant across spectral bands (Lu et al., 2011), primarily due to the spectral correlation of absorption by water vapour, liquid water and ice. In principle, this error can be ameliorated by computing cloud properties separately for each $k$ term, possible if we have fine-scale information on which parts of the spectrum each $k$ term contributes to. As described in section 4.2, this information is requested of participants for
each of their CKD models.

In the final part of CKDMIP, errors in cloudy skies will be estimated. This may be achieved using the clear-sky submissions of the CKDMIP participants, so requiring no additional simulations from them. Firstly, line-by-line cloudy-sky benchmarks are produced. For liquid clouds, Mie calculations have been performed for distributions of droplets at a sufficiently high spectral resolution to resolve variations of refractive index. For ice clouds we use the generalized habit mixture of Baum et al. (2014).
The CKDMIP software is then used to add horizontally homogeneous clouds of varying optical depth to the gas optical depth in the *Evaluation-1* dataset for present-day conditions, and to perform line-by-line calculations. Then the equivalent calculations are performed for the various CKD models, by taking their present-day optical depth files and adding the contribution from clouds. From the information they provide on the spectral contributions to each $k$ term, average cloud properties will be computed for each $k$ term using the appropriate combination of 'thick' and 'thin' averaging (Edwards and Slingo, 1996).
Errors in irradiances and heating rates will then be computed.

A similar procedure would be possible for aerosols, or to quantify errors due to spectrally surface albedo, particularly over snow an vegetation where the variations are largest.

## 5  Evaluation of RRTMG

In this section we demonstrate the CKDMIP approach by using the *Evaluation-1* dataset to evaluate an existing CKD model:
RRTMG. This model is very widely used; for example, Hogan et al. (2017) reported in their survey of seven global NWP models that three used RRTMG for gas optics in both the longwave and the shortwave, and one used it in the longwave only. We evaluate the RRTMG implementation in the ECMWF radiation scheme (Hogan and Bozzo, 2018), which is only slightly modified from the original implementation by Morcrette et al. (2008) and has been found to be indistinguishable from from the gas optics in version 3.9 of RRTMG available from AER.



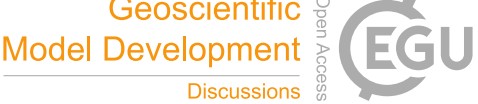

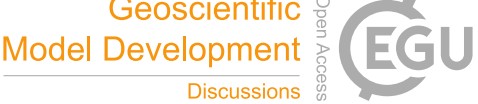

**Figure 5.** Evaluation of longwave irradiances and heating rates from the RRTMG CKD model for the 50 profiles of the *Evaluation-1* dataset with present-day concentrations of the well-mixed greenhouse gases: The left three panels show the downwelling and upwelling irradiances and heating rates from the reference line-by-line calculations. The red lines in the middle three panels show the corresponding bias in the calculation of these quantities from RRTMG. The shaded regions encompass 95% of the errors in the instantaneous profiles (estimated as 1.96 multiplied by the standard deviation of the error). Panels c and f depict instantaneous errors in upwelling TOA and downwelling surface irradiances. The statistics of the comparison are summarized in the lower right, including the root-mean-squared error (RMSE) in heating rate (weighted by the cube-root of pressure) in two ranges of pressure indicated by the horizontal dotted lines in panel h.





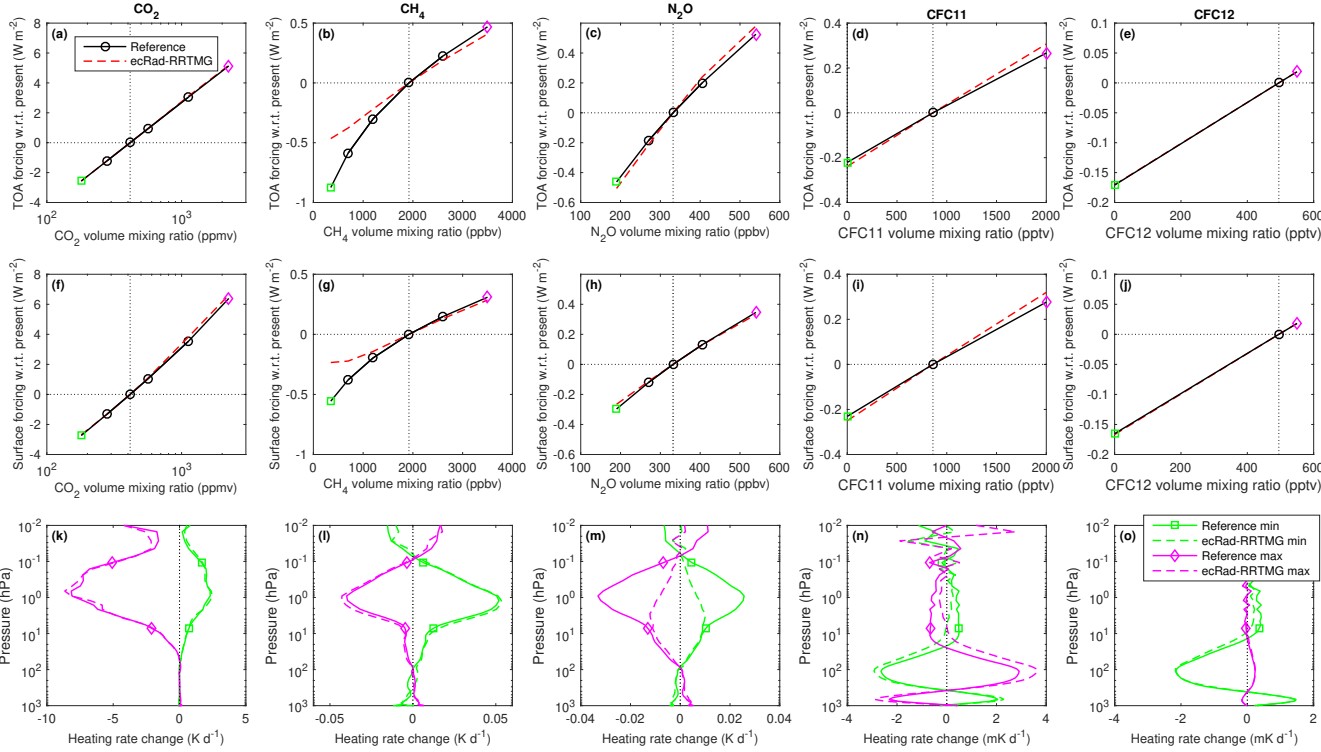

**Figure 6.** Comparison of reference line-by-line and RRTMG calculations of the instantaneous clear-sky radiative forcing from perturbing each of the five well-mixed greenhouse gases from their present-day values, at (top row) top-of-atmosphere and (middle row) surface, averaged over the 50 profiles of the *Evaluation-1* dataset. The bottom row shows the mean change to heating rate resulting from perturbing the concentration of a gas from its present-day value to either the maximum or minimum value in the range shown in Table 2.

Figure 5 evaluates longwave irradiances and heating rates for the present-day scenario described in Table 2. The same radiative transfer algorithm is used for the reference line-by-line calculations and the CKD model: no scattering with four zenith angles per hemisphere. Irradiance errors are almost all within $2 \text{ W m}^{-2}$ at any altitude, and the magnitude of the biases at the surface and TOA are around $0.4 \text{ W m}^{-2}$. Panel h shows that for pressures down to 4 hPa, the heating rate bias is low and the RMSE is only $0.1 \text{ K d}^{-1}$. For lower pressures than this in the upper stratosphere and mesosphere, the heating-rate RMSE is

twice as large and the bias profile exhibits distinct 'wiggles' with pressure. The equivalent plots for the 'preindustrial', 'glacial maximum' and 'future' scenarios may be viewed at the CKDMIP web site, along with an evaluation of the contributions from each of the narrow spectral intervals listed in Table 5.

       Figure 6 uses scenarios 5–22 of Table 2 to evaluate the instantaneous radiative forcing associated with perturbing the concentrations of individual well-mixed greenhouse gases from their present day values. Instantaneous radiative forcing is defined

here as the change to the net (downwelling minus upwelling) irradiance at TOA or the surface, keeping atmospheric and surface temperatures fixed. The radiative forcings have been averaged over the 50 *Evaluation-1* profiles. We see that in general



**Figure 7.** Similar to Fig. 5 but for the shortwave. The reference line-by-line calculations in the left panels are for all 50 atmospheric profiles at five values of the cosine of the solar zenith angle, $\mu_0$. The subsequent evaluation considers all 250 combinations. The blue lines in the middle column of panels show the unmodified RRTMG, and in red after scaling the irradiance profiles in each of the 13 bands in Table 6 to use the same solar irradiance as the reference calculations. Panels c and f compare TOA and surface irradiances for the unmodified and modified versions of RRTMG, with the five clusters of points in each panel corresponding to the five values of $\mu_0$.





RRTMG captures the radiative forcings accurately, including $CO_2$ increased to eight times its preindustrial concentrations. The one exception is the forcing associated with reducing $CH_4$ to 350 ppbv, the magnitude of which is underestimated by around a factor of two. Recent evaluation (not shown) of the new 'parallel' version of RRTMG (RRTMGP; Pincus et al., 2019) has found that this problem has since been fixed, although note that at present RRTMGP uses 256 $k$-terms in the longwave so is more expensive than RRTMG.

Figure 7 evaluates the shortwave irradiance and heating-rate profiles from RRTMG for present-day concentrations of the well-mixed greenhouse gases. RRTMG up to and including version 3.9 uses a solar spectrum from the mid-1990s that has 7–8% more energy in the ultraviolet than the up-to-date Coddington et al. (2016) spectrum used in CKDMIP. This results in an overestimate in solar heating by $O_2$ and $O_3$, which the blue line in Fig. 7h shows to peak at on average 1.5 K d$^{-1}$ at the stratopause. Hogan et al. (2017) reported that the resulting warm bias in the stratospheric climate of the ECMWF model could be reduced by scaling the irradiances in each RRTMG band to match the solar spectral irradiance of Coddington et al. (2016). The red lines and symbols in Fig. 7 show that the effect of doing the same in the 13 bands of Table 6 is to significantly reduce the heating-rate overestimate in the upper atmosphere. Plots evaluating the performance in each of these narrow bands are shown on the CKDMIP web site.

Figure 8 depicts the shortwave radiative forcing resulting from perturbing the concentrations of $CO_2$ and $CH_4$ in the range shown in scenarios 5–14 of Table 2. We see that the radiative forcing is underestimated by 25–45% for both gases, yet the heating rate response is generally good. This implies that there is scope for improvement in the parts of the spectrum where the absorption by $CO_2$ and $CH_4$ is weak but not zero.

The change to the shortwave radiative forcing of perturbing $N_2O$ across its 190–540 ppbv range is around 0.03 W m$^{-2}$ at TOA and 0.15 W m$^{-2}$ at TOA, which is around 10% of that from perturbing $CH_4$ across its 350–3500 ppbv range. Since $N_2O$ is not represented in the shortwave part of RRTMG, a comparison has not been plotted.

# 6 Conclusions

The Correlated K-Distribution Model Intercomparison Project (CKDMIP) is an international collaboration whose aim is to evaluate and improve the treatment of gas optics in the radiation schemes used for weather and climate prediction. In this paper we have described the detailed experimental protocol for CKDMIP, along with the generation of the associated large dataset of gaseous absorption spectra and radiative transfer software.

The nine most radiatively important atmospheric gases in the terrestrial atmosphere have been selected, and via the use of an equivalent concentration of CFC-11 the next 38 most radiatively significant gases are implicitly accounted for. We have found that $N_2$ and $N_2O$ each reduce the daytime surface downwelling shortwave irradiance by of order 0.25 W m$^{-2}$, so ought not to be ignored by shortwave CKD models as they generally are at present.

The primary dataset for evaluation consists of 50 profiles extracted from the ECMWF model, a third of which have been chosen to have extremes of temperature, humidity and ozone. Thirty-four scenarios have been devised for the well-mixed greenhouse gases, intended to span terrestrial concentrations over the last million years and out to the highest concentrations in



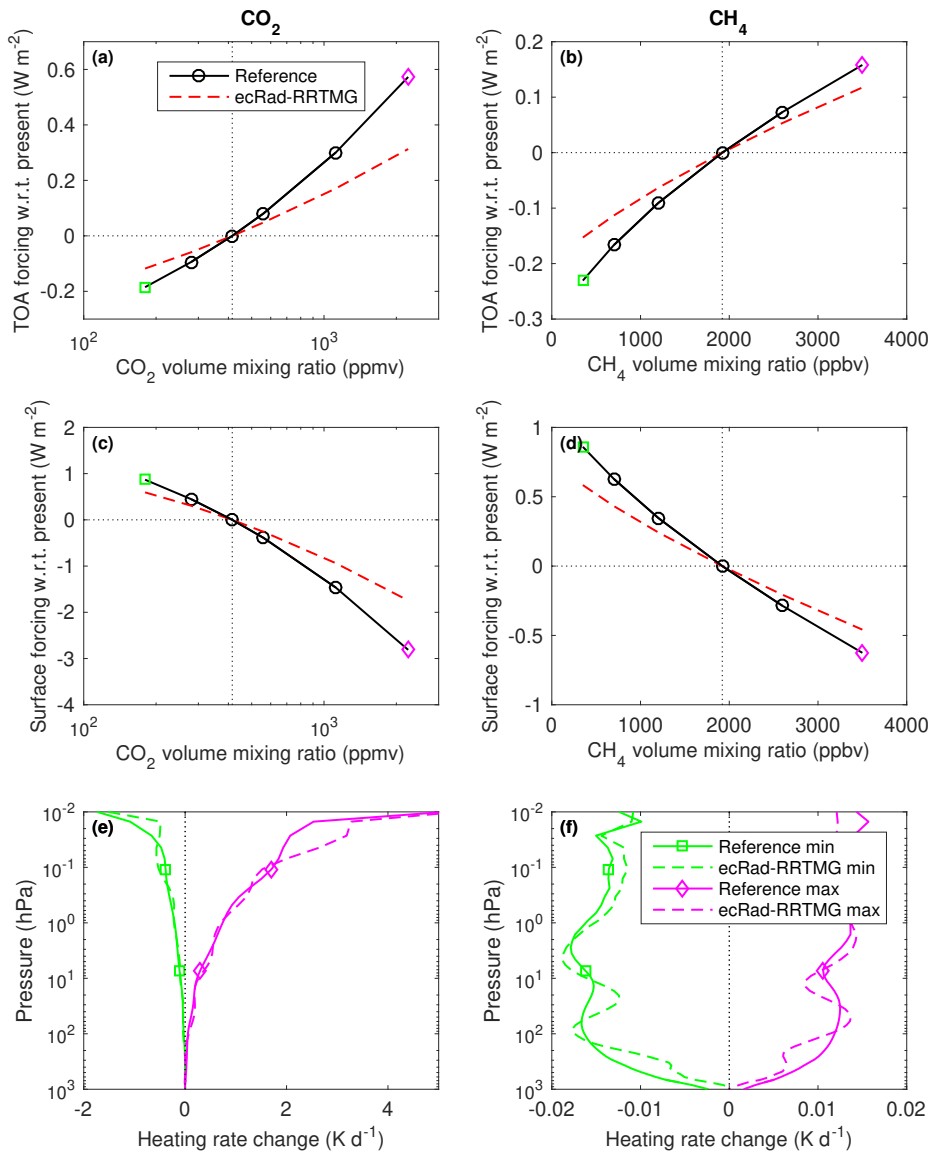

**Figure 8.** As Fig. 6 but for the shortwave radiative forcing by $CO_2$ and $CH_4$. The five solar zenith angles have been averaged so the values shown here represent a daytime average.

any of the CMIP6 projections to the year 2250. We have found that the per-molecule absorption is essentially independent of concentration for all gases except water vapour, which means that line-by-line reference calculations can easily be performed for any scenario (using the CKDMIP software) by scaling the absorption spectra from their present-day values.

We have demonstrated the strengths of the CKDMIP approach by using the dataset and software to evaluate RRTMG, an existing widely used CKD model. This has revealed some particular strengths of RRTMG, such as its ability to estimate the



longwave radiative forcing of the main anthropogenic greenhouse gases in future climate scenarios, but has also uncovered some shortcomings in a few of the bands that will be improved in future versions of RRTMG.

The next step will be to evaluate not just CKD models with fixed numbers of $k$ terms, but CKD *tools* that can generate new CKD models, quantifying how their accuracy varies with the number of $k$ terms (a proxy for the computational cost of an entire radiation scheme). An objective comparison of the performance of different CKD tools will provide crucial insights into

which strategies and approximations yield the most accurate CKD models for a given computational cost. We will also use the submissions by the CKDMIP participants to quantify the errors associated with representing the spectral variation of cloud optical properties, and the extent to which these can be mitigated by using different optical properties for each $k$ term rather than just each band.

In the longer term it is hoped that CKDMIP will stimulate the development of community tools to allow users of radiation

schemes to more easily generate CKD models targeted at specific applications. It could also form the basis for improving the consistency between the broadband irradiance models considered in this paper and the narrowband radiance models used for data assimilation, since the latter are also often trained using line-by-line calculations on a set of training profiles (e.g., Matricardi et al., 2004). Furthermore, while the focus of the CKDMIP dataset is on the terrestrial atmosphere, what is learned during the project should translate easily to radiative transfer on other planets.

*Code and data availability.* The code and technical documentation are available at the project web site http://confluence.ecmwf.int/display/ CKDMIP. The username and password needed to access the FTP site containing the CKDMIP datasets are available on request from the first author.

*Author contributions.* RH carried out the conceptualization, investigation, design of the CKDMIP metholology, writing of the CKDMIP software, data curation and writing of the manuscript. MM generated the raw line-by-line data, contributed to the design of the CKDMIP

methodology, wrote the parts of the manuscript describing the use of LBLRTM and provided critical review of the rest of the manuscript.

*Competing interests.* The authors declare that they have no conflict of interest.

*Acknowledgements.* This paper has benefited from valuable discussions with Eli Mlawer, Robert Pincus, James Manners, Keith Shine, Seiji Kato, David Paynter, Jiangnan Li and Stephen English. Hartwig Deneke is thanked for assistance in comparing various versions of RRTMG with each other.



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
