# Peer review of "Evaluating and improving the treatment of gases in radiation schemes: the Correlated K-Distribution Model Intercomparison Project (CKDMIP)"

_Geoscientific Model Development, 2020_

## Referee Comment (RC1) · Anonymous Referee #1 · 12 Jul 2020

General comments:

The paper describes a valuable intercomparison of modern radiation methods used in NWP and ESM codes with a focus on the underlying formulation of the correlated k-distributions (CKDs). The paper describes the rationale for the MIP protocols, describes how interested groups may participate and contribute their results, and demonstrates the application of these protocols to a particularly widely-used CKD-based code, the Rapid Radiative Transfer Method for GCMs (RRTMG). The topic and nature of this article are both good fits for GMD.

I would recommend acceptance subject to minor revisions – my principal concern is that the protocol is strongly oriented around WMGHGs but should have a much simpler way of attributing errors to the treatment of water vapor, which after all is by far the dominant radiatively active species in both shortwave and longwave.

**Major Comments:**

1. The paper would benefit from citation of recent work in the introduction that demonstrates that there is still substantial room to improve the accuracy of radiative transfer (RT) codes used in climate applications. While not all the codes cited in these works are based on CKD, CKDMIP would not be underway if the accuracy of RT parameterizations relative to line-by-line (LBL) codes was a settled problem. Recent papers that could be cited include Soden et al (2018) that showed that the range of instantaneous RF from doubling $CO_2$ exhibits nearly the same range as it did in prior evaluations including RTMIP (Collins et al, 2006) and going as far back as Cess et al (1993). Other relevant citations include DeAngelis et al (2015) and Fildier and Collins (2015), both of whom showed that inaccuracies in the parameterizations of near-IR absorption by $H_2O$ introduced significant spread in the response of the hydrological cycles simulated by the parent climate models to global warming.

2. Similarly, the paper would benefit from summarizing the codes used by current Earth System Models as configured for the 6th Coupled Model Intercomparison Project (CMIP6). These codes are documented in detail at https://es-doc.org/. Shockingly, there are CMIP6 codes using 30+ year-old shortwave parameterizations and longwave parameterizations based on elaborations of cooling-to-space formulations. Even for codes using CKD in their radiation suite, there are diverse formulations of CKD and this technique is sometimes used for one band (typically longwave) rather than both shortwave and longwave bands.

3. The design of the CKDMIP, particularly the first and second evaluation data sets

(table 1 and 2) containing randomly selected realistic thermodynamics profiles, is going to make it much harder to search for systematic errors associated with the treatment of water vapor than is necessary or feasible. Since water vapor is by far the dominant radiatively active species in both the shortwave and the longwave, and since the literature cited in the first major comment above shows that there is still significant spread in the accuracy of the parameterization of near-IR H2O absorption across the CMIP ensembles, this is my principal concern regarding the design of CKDMIP. It's really important to be able to look at the change in the k distributions and resulting fluxes and heating rates when water vapor **alone** is perturbed. There is a simple fix for this, fortunately, if CKDMIP were to also ask for the exact same set of data from each contributing group for the *idealized* profiles as for the *Evaluation-1* and, ultimately, *Evaluation-2* datasets.

4. Otherwise attribution of errors associated with overlap of WMGHGs and $H_2O$ – e.g., $CO_2$ and $H_2O$, $CH_4$ and $H_2O$, etc. etc – is going to be challenging to put it mildly.

5. The article is curiously silent about the extensive literature on how best to construct CKDs. It should cite existing exhaustive literature on methods, with extensive contributions from the combustion engineering and astronomical communities. A brief search of *JQSRT* should suffice to produce some recent works to cite.

6. Aside from the somewhat empirical objectives of CKDMIP, the authors should consider posing the moderately rhetorical question why isn't there a formal theory for how to, e.g., satisfy cost functions (error limits) with a minimum number of points, that works for the widest possible range of plausible conditions, e.g., mass paths ranging from zero to infinite? After all, CKD is simply a discretization of the Laplace transform of the non-grey Beer-Lambert law – it should be possible to formulate a proper mathematical theory for how to implement this discretization.

7. There is a fair amount of meta-data that should be recorded by each contributing group and it's not clear from the experimental protocols in section 4 that its collection is mandated. Are the provenance of the line data recorded? Assumptions re line width? Variations in continua? Resolution of underlying LBL calculation used to generate k(g)s? For given interval in g, how is k chosen?

**Minor Comments:**

1. Line 406: Why not require mapping to k intervals at the native resolution of the LBL results used by each group? As noted in the article, this would be a vector of approximately 7 million integers for the longwave and 3 million integers for the shortwave, i.e., a trivial but extremely useful volume of data.

2. Line 19: CKD is implemented via a Laplace transform of the Beer-Lambert law followed by reformulation in terms of a cumulative integral – this description in the current version make CKD sound little better than exponential sum fitting.

3. Line 46: Lack of formal theory issue raised in major remarks above.
* * *

---

## Referee Comment (RC2) · Anonymous Referee #2 · 4 Sep 2020

Referee comment on a manuscript:

"Evaluating and improving the treatment of gases in radiation schemes: the Correlated K-Distribution Model Intercomparison Project (CKDMIP)"

by Robin J. Hogan and Marco Matricardi

GENERAL COMMENTS

The manuscript describes CKDMIP project, whose main goal is to improve gas optics in weather and climate models using correlated k-distribution (CKD) method, which is

nowadays a predominant choice for making efficient yet accurate calculation of broadband atmospheric radiative transfer. Desired improvement should arise from an optimal trade-off between efficiency and accuracy, controlled by a choice of spectral bands and associated quadrature points (g-points). First step towards defined goal is an intercomparison of existing CKD models in a unified framework, and understanding how the specific CKD choices affect the quality. For this purpose, a set of atmospheric profiles was created, sampling the range of conditions characterizing current as well as past and anticipated future climate. Nine most radiatively active gases are considered, and the influence of additional 38 trace gases is approximately included via CFC-11-equivalent. For each profile, benchmark radiative fluxes and heating rates were calculated using line-by-line model. Profiles are divided in two indepenent evaluation datasets, so that cross validation of obtained results can be done. Input profiles, benchmark results for evaluation dataset 1, necessary software tools and instructions are available via CKDMIP web page, where the results of intercomparison are being published as well.

Participants to the intercomparison are invited to make either a full submission of the results obtained with their CKD scheme, or a partial submission according to their area of interest. The latter option was enabled given the fact that the full submission is very demanding, requiring generation of 36 configurations (models) of given CKD scheme. Accuracy of results is evaluated with respect to benchmark line-by-line calculations. Efficiency is judged only from the total number of g-points, so it does not reflect the fact how optimally the different CKD schemes are coded. In order to make the intercomparison free of aspects unrelated to CKD method, participants submit clear-sky layer optical depths delivered for each k-term by their CKD models (plus necessary additional information), and the radiative fluxes are then calculated by the CKDMIP software. In this way it is ensured that all submissions will use the same set of approximations in solving the radiative transfer equation. Another interesting aspect is that although participants submit only the clear-sky results, it is possible to make an a posteriori inclusion of clouds. Influence of non-grey clouds on accuracy will thus be

evaluated centrally by the CKDMIP authors.

In final part of the manuscript, CKDMIP approach is applied to a widely used model RRTMG, revealing its strengths and weaknesses. This is very persuasive demonstration of capabilities and usefulness of CKDMIP.

The manuscript is written very clearly, reflecting a thoroughly designed CKDMIP concept. There are hardly any critical comments I can raise, therefore I recommend the manuscript to be accepted for publication, after correcting the typos listed in technical corrections part. I would like to congratulate authors to a nice article, and thank them for investing their time and energy into activity beneficial for a wide community using CKD method. I wish them many CKDMIP submissions and fruitful evaluation of the results, followed by desired improvement of routinely used CKD models.

SPECIFIC COMMENTS

line 304: Even if irrelevant for CKDMIP itself, can you make a comment about relevance of assuming local thermodynamic equilibrium in a pressure range 0.02-4hPa?

TECHNICAL CORRECTIONS

line 52: that it the includes => that it includes

line 104: O2 and N2 have constant mole fractions => O2 and N2 have constant dry air mole fractions

line 153: grid not sufficient => grid is not sufficient

line 214: mean sea level pressure => average mean sea level pressure

lines 243-245: It would be nice to add also information about wavelength ranges. In shortwave spectrum wavelengths are used more frequently than wavenumbers.

Table 4: two significant figures are shown => two significant digits are shown There is missing information that tabulated values are in W/mˆ2.

line 466: spectrally surface albedo => spectrally varying sufrace albedo

line 506: and 0.15W/mˆ2 at TOA => and 0.15W/mˆ2 at the surface

---

## Author Comment (AC1) · 23 Sep 2020

We thank Anonymous Referee #1 for these positive and helpful comments.

1. The revised version of the manuscript will start the introduction with the point that there is substantial room to improve the accuracy of RT codes, citing some of the papers suggested by the reviewer. We will then point out that differences between codes can be due to differences in underlying spectroscopic datasets and/or errors in formulating a CKD model based on a particular spectroscopic dataset. We stress that

CKDMIP is concerned only with the second of these problems; we attempt to sidestep the first problem by providing CKDMIP participants with the spectroscopy that they should use.

2. It is true that it is shocking how old some RT models in CMIP6 are, but the reviewer's Comment 1 indicates that this has already been reviewed in the literature. CKDMIP is not really concerned with the problems associated with old spectroscopy, but rather it uses an up-to-date spectroscopy to focus on the algorithmic challenge of formulating CKD schemes. CKDMIP's contribution to the problem of very old schemes in current climate models will be to improve the efficiency, accuracy and availability of CKD models in future, so hopefully they can be more easily plugged into climate models. This point is made in the conclusions of the revised paper.

3, 4. This ought to be mitigated by the fact that participants are advised to use the same water vapour spectroscopy, but we recognise that in practice not all will. We now make the point in the conclusions that additional cases could be added later in the project if it becomes apparent when the first submissions are received that more are needed. Unfortunately the "Idealized" dataset is not suitable for use directly in radiative transfer because it uses constant water vapour mixing ratios with height, so sensible values near the surface would correspond to unphysical supersaturations higher in the atmosphere.

5. The impact of different assumptions and methods for constructing CKDs on their accuracy will be explored in a later CKDMIP results paper, and this would be the place for a more detailed review of the various methods. However, we have added to the introduction a list of references to the CKD models/tools of the CKDMIP participants so far.

6. This is now mentioned in the conclusions.

7. We do log information from each group, but more in terms of the formulation of their CKD tool rather than the spectroscopy, which is a dataset provided to participants (see https://confluence.ecmwf.int/display/CKDMIP/How+CKD+tools+work). This is now mentioned in section 4.2.

Minor comments

1. Many CKD models, possibly most, do not use a unique mapping from k intervals to parts of the spectrum, but use a separate mapping at different pressures and temperatures. It therefore doesn't seem appropriate to mandate the provision of this data, at least not unless mapping is identified as an important source of differences between CKD models later in the project.

2. The intention is that the text in the first paragraph of the introduction is clear for the widest possible audience, and leads into the discussion of the accuracy-efficiency trade-off. This is best served by talking in terms of reordering and discretization rather than Laplace transforms. If this text implies exponential sum fitting to the reviewer then it simply reflects the fact that the ESFT and CKD share some common features.

3. This is now mentioned in the conclusions.

―――――――――――――――――

---

## Author Comment (AC2) · 23 Sep 2020

We thank Anonymous Referee #2 for these positive and helpful comments.

It was an oversight of our paper not to mention non-LTE effects. The new version of the manuscript will include a brief discussion of this in section 2.2, and also mention in section 3.5 that the LTE assumption is made in the radiative transfer calculations. We will point out that the most well known non-LTE parameterizations for heating rates by Fomichev et al. (JGR 1998) blends between LTE and non-LTE models over a pressure

range of 0.016-0.045 hPa. Therefore in CKDMIP it is appropriate to evaluate the heating rates calculated using the LTE assumptions to a pressure of around 0.02 hPa, but we should always be aware that for the most accurate results in an atmospheric model, these heating rates would need to be blended with those from a non-LTE scheme at pressures lower than around 0.045 hPa.

All the technical corrections have been made in the new manuscript, except for the one indicated for line 214. We don't see why it is necessary to replace "mean" with "average mean" when surely "mean" means "average" already.

————————————————————

---

## Author Response (AR2)

**Hogan and Matricardi: response to referees**

**Response to Referee 1**

We thank Referee 1 for these positive and helpful comments.

**Major comments**

1. The introduction of the revised version of the manuscript now starts with a new paragraph making the point that there is substantial room to improve the accuracy of RT codes, citing some of the papers suggested by the referee. We then cite Collins et al. (2006) who showed that LBL models agree much better with one another than do the fast radiation codes in climate models, indicating that the main problem is not in our knowledge of the spectroscopy, but in the fast models that we use to attempt to mimic expensive LBL calculations. It is this latter problem that CKDMIP is addressing. Later in the introduction we state that by providing participants with common datasets of high spectral resolution gas absorption, we hope to avoid differences due to inconsistent spectroscopy, enabling the results to be interpreted purely in terms of the algorithms used by each CKD tool.

2. It is true that it is shocking how old some RT models in CMIP6 are, but the reviewer's Comment 1 indicates that this has already been reviewed in the literature. CKDMIP is not really concerned with the problems associated with old spectroscopy, but rather it uses an up-to-date spectroscopy to focus on the algorithmic challenge of formulating CKD schemes.  CKDMIP's contribution to the problem of very old schemes in current climate models will be to improve the efficiency, accuracy and availability of CKD models in future, so hopefully they can be more easily plugged into climate models.  This point is made in the conclusions of the revised paper.

3, 4. Regarding inconsistent water vapour spectroscopy, this ought to be mitigated by the fact that participants are advised to use the same water vapour spectroscopy, but we recognise that in practice perhaps not all will. We now make the point at the end of section 3.3, and in the conclusions, that additional datasets and scenarios could be added later in the project if it becomes apparent when the first submissions are received that more are needed. Water vapour is given as an example. Unfortunately the "Idealized" dataset is not suitable for use directly in radiative transfer because it uses constant water vapour mixing ratios with height, so sensible values near the surface would correspond to unphysical supersaturations higher in the atmosphere.

5. The impact of different assumptions and methods for constructing CKDs on their accuracy will be explored in a later CKDMIP results paper, and that would be the place for a more detailed review of the various methods.  However, we have added to the introduction eight additional references, one for each of the CKD tools of the CKDMIP participants signed up so far.

6. We now mention in the conclusions that the evidence compiled in CKDMIP could form a springboard for the development of a more formal theory to underpin CKD tools, such as how to optimally partition g space

7. We do log information from each group, but more in terms of the formulation of their CKD tool rather than the spectroscopy, which is a dataset provided to participants (see https://confluence.ecmwf.int/display/CKDMIP/How+CKD+tools+work).  This is now mentioned in section 4.2.

**Minor comments**

1. Many CKD models, possibly most, do not use a unique mapping from k intervals to parts of the spectrum, but use a separate mapping at different pressures and temperatures. It therefore doesn't seem appropriate to mandate the provision of this data, at least not unless mapping is identified as an important source of differences between CKD models later in the project.

2. The intention is that the text in the introduction is clear for the widest possible audience, and leads into the discussion of the accuracy-efficiency trade-off. This is best served by talking in terms of reordering and discretization rather than Laplace transforms. If this text implies exponential sum fitting to the reviewer then it simply reflects the fact that the ESFT and CKD share some common features.

3. See response to Major Comment 6, above.

**Response to Referee 2**

We thank Referee 2 for these positive and helpful comments.

It was an oversight of our paper not to mention non-LTE effects. The new version of the manuscript includes a discussion of this at the end of section 2.2, and also mention in section 3.5 that the LTE assumption is made in the radiative transfer calculations. We point out that the most well known non-LTE parameterizations for heating rates by Fomichev et al. (JGR 1998) blends between LTE and non-LTE models over a pressure range of 0.016-0.045 hPa. Therefore in CKDMIP it is appropriate to evaluate the heating rates calculated using the LTE assumptions to a pressure of around 0.02 hPa, but we should always be aware that for the most accurate results in an atmospheric model, these heating rates would need to be blended with those from a non-LTE scheme at pressures lower than around 0.045 hPa.

All the technical corrections have been made in the new manuscript, except for the one indicated for line 214. We don't see why it is necessary to replace "mean" with "average mean" when surely "mean" means "average" already.

[revised manuscript text omitted]